# The Immune Landscape and Its Potential for Immunotherapy in Advanced Biliary Tract Cancer

**DOI:** 10.3390/curroncol32010024

**Published:** 2024-12-31

**Authors:** Andry Santoso, Iris Levink, Rille Pihlak, Ian Chau

**Affiliations:** 1Gastrointestinal Unit, The Royal Marsden Hospital, London SW3 6JJ, UK; andry.santoso@rmh.nhs.uk (A.S.); i.levink@erasmusmc.nl (I.L.); 2Department of Gastroenterology and Hepatology, Erasmus MC University Medical Centre, 3015 GD Rotterdam, The Netherlands; 3University Hospitals Sussex NHS Foundation Trust, Brighton BN1 9RW, UK; rille.pihlak@nhs.net

**Keywords:** cholangiocarcinoma, biliary tract cancer, immune microenvironment, immunotherapy

## Abstract

Biliary tract cancers (BTC) are a highly heterogeneous group of cancers at the genomic, epigenetic and molecular levels. The vast majority of patients initially present at an advanced (unresectable) disease stage due to a lack of symptoms and an aggressive tumour biology. Chemotherapy has been the mainstay of treatment in patients with advanced BTC but the survival outcomes and prognosis remain poor. The addition of immune checkpoint inhibitors (ICI) to chemotherapy have shown only a marginal benefit over chemotherapy alone due to the complex tumour immune microenvironment of these cancers. This review appraises our current understanding of the immune landscape of advanced BTC, including emerging transcriptome-based classifications, highlighting the mechanisms of immune evasion and resistance to ICI and their therapeutic implications. It describes the shifting treatment paradigm from traditional chemotherapy to immunotherapy combinations as well as the potential biomarkers for predicting response to ICI.

## 1. Introduction

Biliary tract cancers (BTC) are a highly heterogeneous group of malignant tumours which include true cholangiocarcinoma (CCA), gall bladder carcinoma (GBC) and sometimes ampullary carcinoma. CCAs can be divided into intrahepatic cholangiocarcinoma (iCCA), accounting for 10–20% of CCAs, and extra-hepatic cholangiocarcinoma (eCCA), the latter of which is further divided into peri-hilar carcinoma (pCCA), accounting for 50–60% of CCAs, and distal carcinoma (dCCA), accounting for 20–30% of CCAs [1,2]. These subtypes of BTC differ not only in their anatomical location, but also in their clinical presentation, genetics and molecular drivers [3].

BTCs arise from cholangiocytes, hepatocytes, hepatic stem cells and the peribiliary glands located in the lining of bile ducts [4]. Whilst several genetic mutations, epigenetic alterations and factors in the microenvironment have been uncovered, the full aetiology of BTC and the role of the immune system remains unresolved. It is implicated that tumorigenesis can be triggered by liver diseases characterised by a high inflammatory state. These include infectious diseases (such as liver fluke and hepatitis), cholelithiasis, metabolic-associated fatty liver disease, alcoholic liver disease and liver damage from drugs and smoking [5,6,7]. These conditions have high concentrations of inflammatory mediators that are released as a result of chronic inflammation, which—after a complex interplay between parenchymal cells and stromal cells (immune cells, cancer-associated fibroblasts [CAFs], stellate cells, Kupffer cells and endothelial cells)—can activate cell survival, growth and proliferation [8].

The incidence of BTC varies between countries and regions, depending on the incidence of infectious diseases and the subtype of BTC [5]. Mortality is generally on the rise, with higher rates in countries in South-East Asia (4–6 per 100,000 deaths) as compared to European and American countries (1–2 per 100,000 deaths) [4,6,7]. Surgery remains the only curative treatment option. However, approximately 70% of patients initially present with already advanced (unresectable) disease, due to a lack of adequate early detection tools, challenging anatomical access and often an aggressive tumour biology [9]. Despite advances in BTC knowledge and management, 5-year survival rates remain poor at 7–20% [4].

Doublet chemotherapy of cisplatin and gemcitabine (CisGem) has been the standard first-line treatment for advanced BTC since the ABC-02 trial in 2010 [10]. Immunotherapy or immune checkpoint inhibitors (ICI) are monoclonal antibodies that block the immune inhibiting checkpoints, such as programmed cell death protein 1 (PD-1), programmed death ligand 1 (PD-L1) and cytotoxic T-lymphocyte-associated protein 4 (CTLA-4), and restore the function of the immune system to target cancer cells. The addition of an ICI, pembrolizumab or durvalumab, to CisGem recently demonstrated a survival benefit over chemotherapy alone and has emerged as the new standard of care in advanced BTC [11,12]. However, the magnitude of benefit seen was marginal due to the complex tumour immune microenvironment (TIME) of BTC.

This review appraises our current understanding of the immune landscape of advanced BTC, including emerging transcriptome-based classifications of BTC, highlighting the mechanisms of immune evasion and resistance to ICI, and their therapeutic implications. It describes the shifting treatment paradigm from traditional chemotherapy to immunotherapy combinations as well as the potential biomarkers for predicting response to ICI.

## 2. The Tumour Immune Microenvironment

Cancer cell immune evasion leads to its survival and tumour progression [13]. Evasion cannot be explained by one factor, as it is a complex interplay between cancer cells and different pillars within the microenvironment [14]. For BTC, profound changes in the microenvironment composition are seen during carcinogenesis [9,15]. Cancer immunotherapy aims to modulate the microenvironment to restore an effective immune response and control tumour growth. Here, we will discuss examples of changes in the BTC tumour immune microenvironment (TIME) that lead to immune evasion and lack of immunotherapy efficacy (Figure 1). Understanding these changes may support identification of those patients responsive to immunotherapy and the development of treatment (combinations) that are able to overcome the described hurdles to enable administration in a more heterogeneous target population.

### 2.1. Tumour-Associated Fibrosis

Tumour-associated fibrosis is a major regulator of response to both chemotherapy and immunotherapy. The highly desmoplastic extracellular matrix (ECM), formed by CAFs, in BTC plays an important role in the suppression of the anti-tumor response by T-cells physically blocking immune cell infiltration [16]. These CAFs, which express α-smooth muscle actin [α-SMA] and osteopontin, are one of the most abundant cell types in the tumour microenvironment and contribute to the poor prognosis of the disease [17,18,19]. Direct interactions between CAFs, tumour cells and immune cells cause this immunosuppressive environment by excretion of hepatocyte growth factors (HGF), chemokines such as C-X-C motif chemokine 12 (CXCL12), exosomes and ECM proteins. CAFs are also able to promote transformation of T-lymphocytes into regulatory T-cells (Tregs) and cause myeloid-derived suppressor cells (MDSCs) and tumour-associated macrophages (TAMs) infiltration, which further contributes to immune suppression [20,21,22,23]. Tregs secrete cytokines like IL-10 and TGF-β, which inhibit proliferation of T-cells and dendritic cells, reducing antigen presentation [24]. MDSCs inhibit T-cell activity by releasing arginase-1, reactive oxygen species (ROS) and nitric oxide (NO), further promoting tumour immune evasion [25]. CAFs are a highly heterogeneous group and display considerable plasticity. Although primarily thought of as contributing to carcinogenesis, specific CAF mediators have now been identified to restrict tumour growth instead, making targeting CAFs as a potential therapeutic option challenging [26].

### 2.2. Ineffective Antigen Presentation and Immune ‘Cold’ Tumours

Fundamentally, cells in the biliary tract are immunogenic, as they express tumour associated antigens (TAAs) and neoantigens in their major histocompatibility complex-I (MHC-I) molecules on their cell surface. The immune system can recognise these TAAs and neoantigens as ‘foreign bodies’ [27,28]. However, in case of cancer progression, spontaneous immunity is insufficient to kill cancer cells, causing the cancer cells to evade the immune system.

One implicated mechanism of immune evasion is an ineffective antigen presentation by MHC-I due to defective peptide processing and/or loading machinery. Effective antigen presentation is required for detection and destruction by CD8+ T-cells, which are key effector T-cells responsible for directly killing tumour cells [29]. In addition, CD4+ T-helper 1 (Th1) cells enhance the cytotoxic response by producing cytokines that activate CD8+ T-cells and stimulate other immune components. BTCs have shown to have a decreased (or loss) of MHC-I expression, which is associated to the number of tumour-infiltrating lymphocytes (TILs), tumour stage and overall survival (OS) [30,31,32,33]. Their relation to tumour stage could be explained by the principle of immunoediting, which describes that during early stages of primary tumour development, the tumour remains heterogeneous—because of multiple genetic alterations—harbouring a wide variety of cell clones expressing different MHC-I phenotypes [34]. The present CD8+ T-lymphocytes progressively kill MHC-I-positive cells, leaving the MHC-I-negative ones behind, rendering it impossible to attract new CD8+ T-lymphocytes (‘cold tumour’) [14,35]. Studies describing the spatial distribution of immune cells in BTC indeed show that CD8+ and CD4+ lymphocytes are mainly distributed around the cancer, whereas high numbers of immunosuppressive immune cell populations, such TAMs, Tregs and MDSCs, are in the centre [36,37]. In BTC, this feature is associated to poor response to ICI and poor OS, as well as a high risk of lymph node metastases [36].

### 2.3. Ineffective T-Cell Response and Immune ‘Hot’ Tumours

Conversely, immune ‘hot’ tumours are associated with increased expression of inhibitory immune checkpoint molecules (such as PD-L1, PD-1 and CTLA4) and high density of TILs [38]. These inhibitory checkpoints impede the T-lymphocytes from performing their effector functions (e.g., cytotoxicity). A high number of TILs can be interpreted as a sign of immune recognition and response to immune checkpoint blockade (ICB). However, high TILs can also be associated with ICI resistance, which can be explained by T-cell exhaustion, a mechanism by CD8+ T-cells (that have been primed by antigen-presenting cells) protect themselves from overstimulation and subsequent cell death. These cells have progressive loss of effector functions and self-renewal capacity [39]. Another explanation is the lack or absence of tumour-reactive T-cell receptors (TCRs). Thus, efforts to reactivate intra-tumoral immune response may benefit from simultaneous approaches to delay T-cell exhaustion (for instance, by interacting with TIGIT or TIM-3) and increase the quality of intra-tumoral TCR repertoire [40,41]. Adoptive immunotherapy with chimeric antigen receptor (CAR) T-cells may overcome this. For instance, anti-CD133 CAR T-cells have shown modest efficacy in CD133-expressing BTC cells [42,43].

### 2.4. Tumour-Associated Macrophages

A high density of TAMs in BTC has been associated with poor prognosis [44,45]. TAMs are derived from recruited circulating monocytes (and to lesser extent, resident Kupffer cells) and interact closely with the tumour microenvironment. M1-type TAMs are characterised by CD86 and CD80 and activated by lipopolysaccharide (LPS), tumour necrosis factor-α (TNF-α) and interferon-γ (IFN-γ) and have pro-inflammatory and iNOS-related anti-tumour properties. Conversely, M2 type TAMs, are characterised by CD206 and CD163, and activated by interleukin-4 (IL-4), IL-13 and periostin (secreted by cancer stem cells [CSCs] and CAFs). M2 TAMs reduce the anti-tumour activity of T-lymphocytes, causing angiogenesis via vascular endothelial growth factor (VEGF), interaction with CAFs, expression of PD-L1 and activation of the Wnt/β-catenin signalling pathway [46]. The Wnt/β-catenin signalling pathway has been shown to be highly active in BTC (and related to a high density of TAMs) [47,48]. Various Wnt inhibitors have been developed and are being tested in clinical trials [49,50]. M2-type TAMs also play a role in recruiting other immunosuppressive cells like Tregs and MDSCs and are often the dominant immune cell type in BTC TIME. The interplay between those differentiated to the M1 or M2 TAM phenotype plays a critical role in cancer progression [30,46].

### 2.5. Cancer Stem Cells

Cancer stem cells in BTC have been proposed as drivers of drug resistance and are capable of self-renewal and unlimited replication, leading to disease recurrence [51]. Previous studies have indicated that the immune system can maintain cancer cells or cancer stem cells (e.g., at a metastatic site) in a dormant state of equilibrium with the immune system. These cells are generally characterised by low MHC-I and PD-L-1 expression and release soluble mediators for monocytes recruitment and macrophage priming [51]. It has been proposed that induction of dormancy may be beneficial and should potentially be stimulated to prevent cancer recurrence [52].

## 3. The Effect of Molecular Alterations on the Tumour Immune Microenvironment

Comprehensive genomic profiling of advanced BTCs has identified several targetable molecular alterations that may have an impact on the TIME. The commonly observed mutations differ based on their anatomical subtype. Mutations more frequently observed in iCCA include fibroblast growth factor receptor 2 (*FGFR2*) fusion/rearrangement, isocitrate dehydrogenase (*IDH1/2*) and *BAP1* mutations. This contrasts with eCCA, where we more frequently observe *KRAS*, *TP53 CDKN2A*, *BRCA1* and *SMAD4* and GBC with *HER2* overexpression and amplification [53,54]. These have led to several second-line treatment options for *FGFR2*, *IDH1* and *HER2*, which will be described later.

Analysis of 3067 iCCA cases identified 426 (14%) *IDH1+* and 125 (4%) *IDH2+* patients and highlighted several differences between these and the *IDHwt* patients. The frequency of other targetable alterations, including *FGFR2*, *BRAF* and *ERBB2*, were lower in the *IDH+* population. Other non-targetable alterations, including *TP53*, *CKDN2A/B* and *MTAP,* were also lower in the *IDH+* population. This highlights that *IDH1* is the predominant driver mutation in this population. The *IDH+* TIME appeared to be more immunogenically ‘cold’, with a lower frequency of microsatellite instability (MSI), high tumour mutation burden (TMB) (more than 10 mutations per mega base) and high PD-L1 expression in the population [55]. The authors also identified higher levels of B7-H4 expression in *IDH1+* patients, which is a T-cell co-inhibitory molecule that suppresses CD4+ and CD8+ T-cell proliferation [56]. Increased levels of B7-H4 were similarly observed in patients with *BAP1* loss of function [57]. Improving our understanding of the effect of the other molecular alterations on the TIME would help guide future therapeutic planning strategies.

## 4. Immune Classifications of Biliary Tract Cancer

Several emerging transcriptomic-based classifications of BTCs and their TIMEs have emerged in recent years, with potential therapeutic implications in the future [58]. These classifications have primarily focused on stratifying iCCAs and their TIMEs, but there remains a significant gap in similar analysis for the other BTC subtypes, such as eCCA and GBC. Given that each BTC subtype may have distinct immune profiles and TIME characteristics, it is crucial to extend these immune classifications to include eCCA and GBC for a more comprehensive understanding of their immune landscape.

One of the proposed classifications by Job et al. (2020), based on immune gene expression signatures from 198 surgical iCCA specimens, identified four distinct immune subclasses that are associated with different immune escape mechanisms and vastly different patient outcomes [59]. The most common subtype is the ‘immune-desert’ subclass, occurring in around 45% of iCCAs, and is defined by weak expression of immune and myelofibroblast signatures. This is in line with the concept that the BTC TIME is ‘immune cold’ with a median overall survival of 42 months in this group [59]. The ‘immunogenomic’ subclass, on the other hand, is marked by activation of the inflammatory and immune-checkpoint pathways and has signatures of adaptive and innate immune cells and activated fibroblasts. This was noted in around 10% of iCCAs and had the best median overall survival at 73 months [59]. The other subtypes are ‘myeloid’, occurring in 15% of iCCAs, distinguished by strong expression of monocyte-derived and myeloid gene signatures and the ‘mesenchymal’ subset (25%), characterised by strong expression of fibroblast signatures. The median overall survival times in these two groups are 25 months and 19 months, respectively [59].

Another transcriptomic-based classification, based on the stroma, tumour and immune microenvironment (STIM), combines these microenvironment elements into five distinct iCCA subclasses [60]. These classes are broadly divided into the ‘inflamed’ group, consisting of immune classical and inflammatory stroma, and the ‘non-inflamed’ group, including the desert-like, tumour classical and hepatic stem-like subclasses. The majority of these patients (65%) are in the non-inflamed subtype, which are characterised by low immune infiltration and low stromal infiltration [60]. The ‘desert-like’ class (20%) is enriched with Treg cells, the ‘hepatic stem-like’ class (35%) has abundant M2-like TAMs and characterised by *IDH* and *BAP1* mutations and *FGFR2* fusions, and the ‘tumour classical’ class (10%) is characterised by cell cycle pathway activation [60]. The ‘inflammatory stroma’ class (25%), in addition to the moderate to high immune and stromal infiltration of the ‘immune classical’ class (10%), is characterised with an activated inflammatory stroma, extensive desmoplasia and ECM deposition, T-cell exhaustion and *KRAS* mutations [60].

There are limited data available on how these classifications apply to eCCA and GBC. Research on eCCA has found that this subtype often exhibits a higher degree of immune suppression, characterised by low T-cell infiltration and high levels of immunosuppressive cells such as Tregs [61]. Similarly, GBC has been shown to have a more complex TIME with higher levels of inflammatory cytokines and immune cell infiltration [62]. Despite these findings, comprehensive immune classifications specific to eCCA and GBC have not yet been established, and the immune features of these subtypes remain less well-defined compared to iCCA.

Incorporating immune classifications into clinical practice for all BTC subtypes will require further prospective validation. Clinical trials that stratify patients based on their specific immune subclass, correlating this with targeted therapies aimed at enhancing ICI responses, are essential to establish their role in improving patient outcomes.

## 5. Treatment Paradigm in Advanced Biliary Tract Cancer

### 5.1. First-Line Treatment Options

The standard first-line treatment for advanced BTC has been a chemotherapy combination of cisplatin and gemcitabine (CisGem), since the publication of the ABC-02 trial in 2010 [10]. This phase II/III trial randomised 410 patients with CCA, GBC or ampullary cancer to receive either gemcitabine alone or a CisGem for 6 months. They found improvements in objective response rate (ORR; 81% vs. 72%, *p* = 0.05), progression-free survival (PFS; 8.0 vs. 5.0 months; HR 0.63, *p* < 0.001) and overall survival (OS; 11.7 months vs. 8.1 months; HR 0.64, *p* < 0.001) in the CisGem arm compared to gemcitabine alone (Table 1).

For a decade, various trials aimed to identify a more effective first-line treatment strategy, with minimal success. For instance, a single-arm, phase II trial treated 60 individuals with advanced BTC with a combination of CisGem with Nab-Paclitaxel, which showed an impressive median OS of 19.2 months [64]. This led to the phase III SWOG 1815 trial, which did not show any significant OS benefit for CisGem plus nab-paclitaxel (14 months) compared to CisGem (12.7 months; HR 0.93, *p* = 0.58) [65]. Similarly, another triplet, FOLFIRINOX (5-fluorouracil, irinotecan, oxaliplatin), was not superior to CisGem (PFS 45% vs. 47%, *p* > 0.05) [66].

In 2022, the TOPAZ-1 trial exhibited improved efficacy of adding durvalumab, an ICI targeting PD-L1, to CisGem in patients with advanced or metastatic BTC [11]. The combination of CisGem with durvalumab led to an improved ORR (27% vs. 19%; odds ratio [OR] 1.6, *p* < 0.05), PFS (7.2 vs. 5.7 months; HR = 0.75, *p* = 0.001) and (an updated) median OS (12.9 vs. 11.3 months; HR 0.76), as compared to CisGem alone (Table 1) [63]. Kaplan–Meier curves were increasingly divergent over time with a durable survival benefit of durvalumab at 18 months (34.8% vs. 24.1%) and 24 months (23.6% vs. 11.5%) [11,63]. Interestingly, no association between PD-L1 expression and response was seen. Even tumours with absent PD-L1 expression had similar survival to those with a PD-L1 tumour area positivity (TAP) score ≥ 1% [11]. The addition of durvalumab to CisGem did not increase the risk of grade 3–4 toxicities (76% vs. 78%). Based on these results, the combination of CisGem plus durvalumab became the new standard of care for the first-line treatment of advanced BTC.

Similar to TOPAZ-1, KEYNOTE-96 randomly assigned 1069 patients to CisGem plus pembrolizumab (anti-PD-1 monoclonal antibody) or CisGem [12]. CisGem plus pembrolizumab exhibited an improved OS (12.7 months) as compared to CisGem (10.9 months, HR 0.83, one-sided *p* = 0.003). The ORR was not different between the groups (29% vs. 29%), yet the duration of response was longer in the CisGem plus pembrolizumab group (9.7 vs. 6.9 months), and the grade 3–4 adverse event rates were not different (70% vs. 69%; Table 1). The combination of CisGem and pembrolizumab was recently approved for use in the first-line setting of advanced BTC in Europe and US.

The phase II IMMUCHEC trial, presented in 2022, showed no significant benefit of the addition of tremelimumab, a CTLA-4 inhibitor, as a second ICI to various combinations of durvalumab and/or gemcitabine and/or cisplatin of advanced treatment-naive BTC [67]. Importantly, it showed that removing cisplatin from the combination led to worse OS and highlighted that the combination of CisGem is an important backbone for the first-line treatment of advanced BTC [67].

A smaller phase II BilT-01 trial also investigated the efficacy of nivolumab (anti-PD-1 monoclonal antibody) and CisGem, as compared to nivolumab and ipilimumab (anti-CTLA-4 monoclonal antibody) [68]. This trial showed a 6-month PFS of 59.4% in CisGem plus nivolumab and 21.2% for nivolumab plus ipilimumab, not meeting the trial primary endpoint. There was also no difference in PFS (6.6 vs. 3.9 months, *p* = 0.077) and OS (10.6 vs. 8.2 months, *p* = 0.61) in CisGem plus nivolumab, compared to ipilimumab plus nivolumab, respectively.

The phase II IMbrave151 trial investigated the addition of bevacizumab (bev) (anti-VEGF monoclonal antibody) to CisGem + atezolizumab (atezo) (anti-PD-L1 monoclonal antibody). The addition of bevacizumab did not significantly improve PFS (8.4 vs. 7.9 months), 6-month PFS rates (78% vs. 63%) or ORR (24% vs. 25%). Further studies to elucidate the use of VEGF inhibitor in BTC are needed.

### 5.2. Second-Line Treatment Options

#### 5.2.1. Conventional Chemotherapy

The ABC-06 trial (2021) was the first phase III randomised controlled trial to show the benefit of giving second-line chemotherapy in patients with advanced BTC [69]. This trial randomised 162 patients with advanced CCA, GBC and ampullary cancer between 5-FU plus oxaliplatin (FOLFOX) chemotherapy and active symptom control (ASC) in the UK. This trial showed longer median OS in the chemotherapy arm (6.2 months), as compared to ASC (5.3 months, *p* = 0.03), and led to the implementation of FOLFOX as the new standard second-line treatment for BTC. The trial also highlighted the importance of ASC in these patients, as it exceeded the expected survival of the control arm (Table 2).

Beyond FOLFOX chemotherapy, other second-line (or later) treatment options like 5-FU plus irinotecan (FOLFIRI) and/or nano-liposomal irinotecan (Nal-IRI) could be considered, but current evidence is based on smaller and/or Asian cohorts only, and the benefit has not yet been translated to the European population [70,71,77,78,79]. For instance, the NIFTY (phase IIb) trial from South Korea (*n* = 193) exhibited a longer PFS for those treated with 5-FU (and leucovorin) plus Nal-IRI (7.1 months), as compared to 5FU (and leucovorin) alone (1.4 months; HR 0.56, *p* = 0.0019) [70]. Conversely, the NALIRICC phase II trial from Germany (*n* = 100) did not show a PFS benefit (Table 2) [71]. Median overall survival was also noted to be longer in the control arm at 8.2 months as compared to 6.9 months in the 5-FU (and leucovorin) plus Nal-IRI arm.

#### 5.2.2. Targeted Therapy

Molecular profiling has become increasingly common in all cancer types, including BTC. Up to 40% of patients with BTC have been found to harbour potentially targetable alterations [80]. These are most prominent in iCCA and have led to (mostly second-line) treatment options in patients with these alterations. Therefore, subtyping of BTC has undergone a paradigm switch from characterisation based on anatomical location to molecular subtyping [81].

*FGFR2* fusions and rearrangements are present in 10–20% of patients with iCCA and only up to around 5% of eCCAs [80,82]. Currently, multiple FGFR-inhibitors (pemigatinib and futibatinib) have been approved by the FDA and/or EMA for second-line treatment of advanced BTC. Pemigatinib was the first to be approved due to the results from the phase II FIGHT-202 trial, which showed an ORR of 36%, median duration of response (DOR) of 7.5 months, PFS of 6.9 months and OS of 21.1 months in patients treated with pemigatinib [72]. FOENIX-CCA2, presented in 2020, showed a PFS of 9.0 months and OS of 21.7 months for 103 patients treated with futibatinib (Table 2) [73]. Several other FGFR inhibitors are currently being investigated in iCCA as first- and second-line FGFR inhibitors in this subgroup.

Similarly, *IDH1* mutations have been found in around 15% of patients with iCCA [83,84,85]. The phase III ClarIDHy trial showed, in patients with *IDH1* mutations, improvements in PFS (2.7 months vs. 1.4 months, HR 0.37, one-sided *p* < 0.0001) and OS (10.8 vs. 9.7 months, HR 0.69, *p* = 0.06) by IDH1 inhibitor ivosidenib, as compared to placebo, and led to FDA and EMA approval of ivosidenib in the second-line setting [74].

*HER2* overpexression can be found in approximately 15% of all BTCs with significant heterogeneity of their levels of expression based on their anatomical locations. *HER2* positivity, defined as IHC 3+ or 2+ and fluorescence in situ hybridisation (FISH)-positive, was more commonly found in GBC (31.3% of patients) and dCCA (18.5% of patients). This is in comparison to only 3% positivity in both iCCA and pCCA [86]. Zanidatamab, a HER2-targeting bispecific antibody that binds to two distinct, non-overlapping HER2 domains, was evaluated in the HERIZON-BTC-01 study for patients with HER2-positive metastatic BTC. This showed a notable ORR of 41%, a median DOR of 14.9 months and median OS of 15.5 months, which led to its very recent approval by the FDA [75]. The phase 2 HERB trial that investigated trastuzumab deruxtecan (T-DXd), an antibody–drug conjugate composed of an anti-HER2 antibody and a topoisomerase I inhibitor in unresectable or recurrent BTC, found an ORR of 36.4%, PFS of 4.4 months and OS of 7.1 months [87]. The DESTINY-PanTumor02 basket trial, which explored T-DXd in multiple HER2-positive solid tumours, recently led to tumour-agnostic approval by the FDA in patients with metastatic HER2 IHC 3+ solid tumours, including BTC, who received prior treatment and no alternative treatment options. This trial included 41 patients with HER2+ (IHC 2+ and 3+) advanced BTC with an ORR of 22%, PFS of 4.6 months and OS of 7 months. Patients with HER2 IHC 3+ BTC have a significantly higher ORR, PFS and OS compared to the overall BTC population at 56.3%, 7.4 months and 12.4 months, respectively (Table 2) [76].

Other molecular subgroups in BTC that are currently being investigated in various trials with early promising results include *BRAF* V600E mutations (5–7%) and *NTRK* fusions (<1%) [83,84,85,88,89,90]. There are multiple clinical trials currently ongoing that also combine ICI, targeted treatments or chemotherapy, with results eagerly awaited.

#### 5.2.3. Immune Checkpoint Inhibitors

Currently, results of single-agent ICI, in unselected patients with advanced BTC, have been limited. However, ICI has been licensed in two tumour-agnostic subgroups: patients with mismatch repair (MMR)-deficient or MSI-high tumours and those with a high TMB [91,92,93]. These subgroups are rare in advanced BTC, occurring in <5% of patients. In the KEYNOTE-158 prospective biomarker analysis, there were 0 patients with advanced BTC with high TMB [94]. Median TMB scores among responders and non-responders in the biliary cohort were 3.15 and 2.52, respectively.

Thus far, ICI-monotherapy has not shown efficacy in BTC. Piha-Paul et al. (2020) evaluated the response to pembrolizumab in patients with advanced BTC from the KEYNOTE-158 (phase II; *n* = 104; 61 patients with PD-L1 ≥ 1%) and KEYNOTE-028 (phase Ib; *n* = 24; all PD-L1 ≥ 1%) trials [95]. It showed ORRs of 5.8 and 13%, PFSs of 1.8 and 2.0 months and OS durations of 5.7 and 7.2 months, respectively (Table 3). Similarly, Kang et al. (2020) showed, in patients with (any) positive PD-L1 staining, an ORR of 10%, PFS of 1.5 months and OS of 4.3 months to pembrolizumab, in patients with advanced BTC, who also previously received two or three lines of treatment [96]. Remarkably, Kim et al. (2020) showed an ORR of 22%, median PFS of 3.7 months and OS of 14.2 months in a single-arm multicentre phase II study of 54 patients with BTC treated with nivolumab monotherapy; these results are clearly better than those of other ICI monotherapy trials [97]. Of note, patients in these studies were treated with ICI monotherapy after progression to one or more prior line(s) of chemotherapy.

Combinations of ICI that target different immune checkpoints have been investigated in BTC in the second-line setting. A phase I trial by Doki et al. (2022) displayed acceptable safety profiles of both durvalumab as monotherapy and in combination with tremelimumab (CTLA-4 inhibitor) in an Asian population with advanced BTC, oesophageal and head-and-neck cancer [98]. In the BTC subgroup (*n* = 42), the durvalumab + tremelimumab combination therapy showed a higher ORR (10.8%) than durvalumab alone (4.8%; Table 3). Klein et al. (2020) presented a phase II non-randomised study with advanced rare cancers and showed, in the subgroup analysis of 39 patients with mostly previously treated advanced BTC, that a combination of nivolumab and ipilimumab reached an ORR of 23%, PFS of 2.9 months and OS of 5.7 months [99]. Interestingly, most responses in the study were limited to patients with iCCA and GBC. A more recent and larger trial to further evaluate combination ICI therapy recruited 60 patients with advanced iCCA or GBC to receive nivolumab and ipilimumab [100]. The majority of these patients were pre-treated with one line of treatment (*n* = 47), including 12 who had previous ICI durvalumab. Response rates were low at 15% in the overall population, which was mainly contributed to by the iCCA population that had a 6% response rate. The GBC response rate was 32%, suggesting that this subtype of advanced BTC may be more immunogenic and responsive to ICI. The combination of T-DXd and pembrolizumab with or without chemotherapy could be a potential option for HER2+ GBC. T-DXd and chemotherapy was found to be tolerable and feasible in the gastric cancer population [101].

## 6. Biomarkers to Predict Immunotherapy Response

### 6.1. PD-L1 Expression

PD-L1 is an immune checkpoint that can be expressed at the surface of macrophages, lymphocytes and cancer cells. PD-L1 binding to the T-cell surface receptor PD-1 suppresses auto-immunity by down-regulating T-cell activation, signalling and apoptosis. PD-L1 expression can be determined either by the percentage of PD-L1 positive tumour cells alone (Tumour Proportion Score [TPS]) or both tumour and immune cells (Combined Positive Score [CPS]) out of the total number of viable tumour cells [102]. Clinical trials with durvalumab, including TOPAZ-1, used the TAP score to assess PD-L1 positivity [11]. This is a visual-based estimation of the space occupied by both stained PD-L1 tumour and immune cells in a given tumour area [103]. However, none of these scoring systems have reliably predicted ICI response in BTC.

In BTC, PD-L1 expression was detected in 8.1% of iCCA, 6.9% of eCCA and 8.0% of GBC tissues [54]. Whilst PD-L1 has emerged as a general biomarker for response to ICI in other cancer types, its predictive value in BTC is less clear [104,105,106]. Studies such as KEYNOTE-028, KEYNOTE-158, TOPAZ-1 and KEYNOTE-966 showed that PD-L1 expression did not correlate with ICI response [11,12,95]. Some studies such as those by Kim et al. and Kang suggested that higher TPS scores may be associated with better ICI response as compared to lower TPS scores [96,97]. These findings are inconsistent and point to the complexity of the TIME in BTC.

A significant limitation of PD-L1 expression as a predictive biomarker is the spatial variability of PD-L1 within tumours. The distribution of PD-L1 expression can be heterogeneous and dynamic, with varying levels across different areas of the tumour. This variability may lead to inconsistent results when assessing PD-L1 positivity using traditional methods [107]. Moreover, the TAP scoring system, being visual-based, may not capture the complexity of the TIME, as it does not account for dynamic interactions between tumour and immune cells across different regions of the tumour [108]. To address these challenges, emerging technologies such as spatial transcriptomics and multiplex immunohistochemistry hold promise. These methods allow for a more comprehensive assessment of PD-L1 expression, considering spatial and cellular contexts, which may provide a more accurate reflection of its role in immune evasion [109,110].

In some cases, PD-L1 expression on resting immune cells in CCA has been linked to worse clinical outcomes in patients who did not receive ICI treatment, but it was associated with better efficacy in patients who did undergo ICI therapy [111,112,113,114]. This suggests that immune cell context and prior exposure to immune modulation may influence the predictive value of PD-L1 expression in BTC.

### 6.2. MMR Deficiency, MSI-High and Tumour Mutational Burden

In the context of ICI, MSI-high and MMR deficiency are well established biomarkers with tumour-agnostic approval for single-agent PD-1 inhibitors like pembrolizumab and nivolumab [91,92]. In the United States, these PD-1 inhibitors are also approved for patients with high TMB, as these tumours tend to harbour numerous somatic mutations, leading to increased neoantigen presentation and enhanced susceptibility of immune checkpoint inhibition [115]. However, BTCs generally have low TMB (median, 1.9/Mb; range, 0.5–11.0/Mb), with only a small percentage of cases exhibiting high TMB—approximately 2% in eCCAs and 3.5% in iCCAs and GBCs [54,116,117]. This is primarily due to the specific genomic landscape of BTCs, which are driven by mutations such as *IDH1/2*, *KRAS* and *TP53*, rather than the accumulation of numerous somatic mutations typically seen in high-TMB cancers [118,119]. Similarly, MMR deficiency and MSI-High are rare in BTC, with only 1% of GBCs and 2.5% of iCCAs showing these features [54]. As a result, the overall mutational burden remains low in BTC, limiting the predictive value of TMB and MSI-High for ICI response.

In the KEYNOTE-158 trial, none of the patients with advanced BTC had high TMB, though 22 patients with MMR deficiency/MSI-High were identified. These patients showed a promising overall response rate of 40.9% with a median PFS of 4.2 months and median OS of 24.3 months [120]. These findings underscore the potential benefit of targeting MMR-deficient/MSI-High BTCs with ICI, though the rarity of this subtype limits its broader applicability.

The low frequency of MMR deficiency/MSI-High and high TMB in BTC highlights the need for alternative biomarkers that may better reflect the immune landscape of these tumours. Given the rarity of these genomic features in BTC, other characteristics such as chromosomal instability and epigenetic silencing could serve as potential alternatives. These features have been implicated in tumorigenesis and immune escape in other cancers and may warrant further exploration in BTC as predictive biomarkers for ICI response [121,122]. Further investigation into these mechanisms is needed to assess their potential utility in guiding ICI treatment strategies for BTC patients.

### 6.3. Other Predictive Factors

In other cancer types, primary resistance to ICI was less frequently observed in the first-line or neoadjuvant setting but was more widely reported in metastatic disease [123,124]. In the KEYNOTE-966 trial, patients with locally advanced BTCs experienced a greater survival benefit from the combination of CisGem plus pembrolizumab than those with metastatic disease [12]. Patients undergoing second-line treatment may have a heavier tumour burden, poorer performance status and impaired immune system after treatment with conventional chemotherapy. These factors can significantly impair the effectiveness of ICI and complicate the identification of reliable predictive biomarkers.

## 7. Therapeutic Strategies to Overcome Immunotherapy Resistance

BTC is characterised by a highly immunosuppressive TIME that contributes to resistance against ICI and necessitates novel therapeutic strategies to enhance response rates. Stromal-targeting agents, such as colony-stimulating factor-1 receptor (CSF1R) inhibitors or Wnt/β-catenin signalling inhibitors, offer promising approaches to remodel the TIME. CSF1R inhibitors can shift TAMs from an immunosuppressive M2 phenotype to a pro-inflammatory M1 phenotype. This reprogramming enhances the recruitment of cytotoxic T-cells and increases the anti-tumour immune response [125]. Additionally, Wnt/β-catenin signalling is associated with immune exclusion, whereby tumours limit immune cell infiltration [126]. By inhibiting this pathway, Wnt/β-catenin inhibitors may enhance the infiltration of T-cells into the tumour site [127]. When combined with ICI, these stromal-targeting agents may help convert immunogenic cold tumours into immunogenic hot ones, potentially improving treatment outcomes by facilitating sustained immune activation.

Therapies that induce immunogenic cell death represents another promising strategy to modify the TIME and enhance ICI response. Immunogenic cell death is a form of programmed cell death that promotes the release of TAAs, which stimulates and releases pro-inflammatory cytokines and chemokines and subsequently dendritic cells and cytotoxic T-cells [128]. Radiotherapy, oncolytic viruses and certain chemotherapies have been shown to induce immunogenic cell death in various cancers, including BTC [129,130]. For instance, radiotherapy induces localised tumour cell death, which can trigger systemic immune responses and lead to the abscopal effect—where distant, non-irradiated tumours also shrink [131]. In a case series of stereotactic body radiation therapy (SBRT) combined with ICI in the treatment of CCA, all four patients demonstrated a clinical response [132]. This highlights the potential of combining ICI with therapies that induce immunogenic cell death to improve outcomes in BTC.

Adaptive T-cell therapies, including chimeric antigen receptor (CAR)-T cells and T-cell receptor (TCR)-based therapies, are emerging as innovative approaches to enhance immune responses in BTC. CAR-T cell therapy involves engineering a patient’s T-cells to express a receptor that specifically recognises and binds to tumour-specific neoantigens, such as glypican-3 (GPC3) in BTC [133]. Similarly, TCR-based therapies involve modifying T-cells to recognise tumour-specific neoantigens, which are generated by mutations in the tumour genome [134]. The combination of adaptive T-cell therapies with ICI could help bypass immune evasion mechanisms and enhance the efficacy of ICI in BTC by increasing the number of active T-cells within the TIME. Several clinical trials are currently exploring adaptive T-cell therapies in the setting of BTC.

Therapeutic vaccines targeting BTC-specific TAAs or neoantigens are another strategy to prime the immune system and complement ICI. These vaccines stimulate the patient’s immune system to recognise and attack tumour cells expressing specific antigens, thereby increasing the pool of tumour-specific T-cells and potentially enhancing the immune response [135]. Additionally, modulating the gut microbiome through probiotics or faecal microbiota transplantation may influence systemic immunity and improve ICI responses, as observed in other malignancies [136].

## 8. Conclusions

BTC represents a group of cancers located in or near the bile ducts that is characterised by resistance to chemotherapy and immune evasion. The majority of BTCs can be best distinguished as immunogenic cold cancers, showing a complex fibrotic microenvironment not responding to ICI monotherapy and doublet therapy. However, a subgroup seems to respond to the combination of chemotherapy and ICI. More research is needed to identify predictive biomarkers that provide the selection of patients likely to respond to this combination. Future research focusing on turning the variety of immunogenic cold tumours into more immunogenic hot tumours is required. Transcriptomic-based immune classification of BTC has yet to show clinical utility but is being investigated further.

## Figures and Tables

**Figure 1 curroncol-32-00024-f001:**
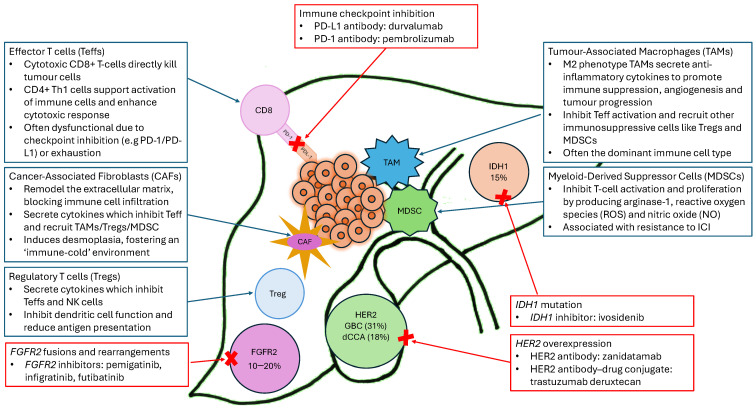
The immunosuppressive microenvironment of biliary tract cancer with its key components, including the common targetable mutations and approved treatment options.

**Table 1 curroncol-32-00024-t001:** Selection of clinical trials investigating first-line treatments of advanced BTC.

Trial	*n*	Treatment Arms	Population	ORR	PFS	OS
ABC-02 (2010) Phase II–III [10] NCT00262769	410	CisGem vs. gemcitabine	Locally advanced or metastatic CCA, GBC and ampullary cancer	81% vs. 72% (*p* = 0.05)	8.0 vs. 5.0 months (HR 0.63, *p* < 0.001)	11.7 vs. 8.1 months (HR 0.64, *p* < 0.001)
TOPAZ-1 (2022) Phase III [63] NCT03875235	685	CisGem ± durvalumab	Locally advanced or metastatic CCA, GBC	27% vs. 19% (*p* < 0.05)	7.2 vs. 5.7 months (HR 0.75, *p* = 0.001)	12.9 vs. 11.3 months (HR 0.76) (updated)
KEYNOTE-966 (2023) Phase III [12] NCT04003636	1564	CisGem ± pembrolizumab	Locally advanced or metastatic CCA and GBC	29% vs. 29%	6.5 vs. 5.6 months (HR 0.86, *p* = 0.023 *)	12.7 vs. 10.9 months (HR 0.83, *p* = 0.003)

* one-sided *p*-value. ORR = objective response rate, PFS = progression-free survival, OS = overall survival, CisGem = gemcitabine plus cisplatin, CCA = cholangiocarcinoma, GBC = gall bladder cancer, HR = hazard ratio, BTC = biliary tract cancer.

**Table 2 curroncol-32-00024-t002:** Selection of clinical trials investigating second-line treatments of BTC.

Trial	*n*	Treatment Arms	Population	ORR	PFS	OS
Conventional chemotherapy						
ABC-06 (2021) phase III [69] NCT01926236	162	FOLFOX vs. ASC	Locally advanced or metastatic CCA, GBC, ampullary cancer	5% (vs. NA)	4.0 months (vs. *NA*)	6.2 vs. 5.3 months (HR 0.69, *p* = 0.03)
NIFTY (2021) Phase IIb [70] NCT03524508	174	5-FU (+leucovorin) + Nal-IRI vs. 5-FU (+leucovorin)	Metastatic CCA and GBC	15% vs. 6%	7.1 vs. 1.4 months (HR 0.56, *p* = 0.002)	8.6 vs. 5.5 months (HR 0.68, *p* = 0.002)
NALIRICC (2023) Phase II [71] NCT03043547	100	5-FU (+leucovorin) + Nal-IRI vs. 5-FU (+leucovorin)	Metastatic BTC	14% vs. 4%	2.6 vs. 2.3 months (HR 0.87)	6.9 vs. 8.2 months (HR 1.08)
Targeted therapy						
FIGHT-202 (2020) Phase II [72] NCT02924376	107	Pemigatinib (single-arm)	Previously treated, advanced CCA with *FGFR2* fusion or rearrangement	ORR 36% (DOR 7.5 months)	6.9 months	21.1 months
FOENIX-CCA2 (2023) Phase II [73] NCT02052778	103	Futibatinib (single-arm)	Advanced iCCA with *FGFR2* fusion or rearrangement	42% (DOR 9.7 months)	9.0 months	21.7 months
ClarlDHy (2020) Phase III [74] NCT02989857	230	Ivosidenib vs. placebo	Advanced, *IDH1*-mutant CCA	3% vs. 0%	2.7 vs. 1.4 months (HR 0.37, one-sided *p* < 0.0001)	10.8 vs. 9.7 months (HR 0.79, one-sided *p* = 0.09)
HERIZON-BTC-01 (2023) [75] NCT04466891	87	Zanidatamab (single-arm)	Previously treated *HER2*-amplified locally advanced, or metastatic BTC	41%	NA	15.5 months
DESTINY-PanTumor02 (2024) Phase II [76] NCT04482309	41 16 IHC3+	Trastuzumab Deruxtecan	Locally advanced, unresectable or metastatic *HER2*+ BTC (IHC2+/3+)	26.8% overall 56.3% in IHC3+	4.6 months overall 7.4 months in IHC3+	7.0 months overall 12.4 months in IHC3+

ORR = objective response rate, DOR = duration of response, PFS = progression-free survival, OS = overall survival, 5-FU = Fluorouracil, Nal-IRI = nano-liposomal irinotecan, CCA = cholangiocarcinoma, GBC = gall bladder cancer, NA = not available, HR = hazard ratio, ASC = active symptom control, BTC = biliary tract cancer, *FGFR2* = fibroblast growth factor receptor 2.

**Table 3 curroncol-32-00024-t003:** Selection of clinical trials investigating second-line immunotherapies in advanced BTC.

Study	*n*	Treatment	Population	PD-L1 (Pos/Neg)	ORR	Median PFS	Median OS
Immunotherapy monotherapy							
Keynote-158 (2020) Phase II [95] NCT02628067	104 (45% 3L)	Pembrolizumab	Advanced CCA and GBC	61/34	5.8% (DOR NA)	2.0 months	7.4 months
Keynote-028 (2020) Phase Ib [95] NCT02054806	24 (50% 3L+)	Pembrolizumab	Advanced CCA and GBC	24/0 (≥1%)	13% (DOR NA)	1.8 months	5.7 months
Kang et al. (2020) [96] NCT03695952	40	Pembrolizumab	Advanced CCA and GBC	40/0	10% (DOR 6.3 months)	1.5 months	4.3 months
Kim et al. (2020) Phase II [97] NCT02829918	54 (56% 2L)	Nivolumab	Advanced CCA and GBC	18/24	22% (DOR NA)	3.7 months	14.2 months
Doki et al. (2022) Phase I [98] NCT01938612	42	Durvalumab	Advanced CCA and GBC	19/23	4.8% (DOR 9.7 months)	1.5 months	8.1 months
Immunotherapy combinations							
CA209-538 Phase II [99] NCT02923934	39	Nivolumab + ipilimumab	Advanced CCA and GBC	NR	23%	2.9 months	5.7 months
Doki et al. (2022) Phase I [98] NCT01938612	42	Durvalumab + tremelimumab	Advanced CCA and GBC	18/47	10.8% (DOR 8.4 months)	2.9 months	10.1 months
MoST-CIRCUIT (2024) Phase II [100] NCT04969887	60	Nivolumab + ipilimumab	Advanced iCCA and GBC	NR	15% (6% in iCCA, 32% in GBC)	6-month PFS 22.6%	6.7 months

PD-L1 expression in tumour cells was <1% in 82% and ≥1% in 18%; PD-L1 expression in tumour-associated immune cells was <1% in 35% and ≥1% in 65%. ORR = objective response rate, DOR = duration of response, PFS = progression-free survival, OS = overall survival, CCA = cholangiocarcinoma, GBC = gall bladder cancer, NA = not available, 2L = 2nd line, 3L = 3rd line.

## Data Availability

No new data were created or analysed in this study. Data sharing is not applicable to this article.

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
