# Peer review of "The Immune Landscape and Its Potential for Immunotherapy in Advanced Biliary Tract Cancer"

_curroncol, 2024, doi:10.3390/curroncol32010024_

Round 1
Reviewer 1 Report
Comments and Suggestions for Authors
This paper provides a comprehensive and insightful review of the immune landscape of biliary tract cancers (BTCs), focusing on mechanisms of immune evasion, resistance to immune checkpoint inhibitors (ICIs), and potential therapeutic approaches. It is well-written, thoroughly researched, and logically organized, making a valuable contribution to the field of BTC research. The authors have effectively summarized recent advances, including transcriptome-based classifications, tumor microenvironment characteristics, and emerging treatment paradigms that integrate chemotherapy with immunotherapy.
Major:
- Discussion of Biomarkers: While the paper provides an excellent summary of commonly studied biomarkers for immunotherapy (e.g., PD-L1, TMB, MSI-H), it does not sufficiently address the challenges and limitations of these biomarkers in BTC. For example:
- PD-L1 Expression: The lack of correlation between PD-L1 expression and ICI response in BTC could be elaborated. Issues such as spatial variability of PD-L1 within tumors or the limitations of TAP scoring in capturing the complexity of the tumor immune microenvironment should be discussed. Briefly mentioning emerging approaches, such as spatial transcriptomics or multiplex immunohistochemistry, as potential solutions could add value.
- TMB and MSI-H: The paper could elaborate on why these biomarkers are rare in BTC and discuss alternative genomic or epigenomic features (e.g., chromosomal instability or epigenetic silencing) that may be relevant for predicting response to ICIs.
- Therapeutic Strategies to Overcome ICI Resistance: The authors effectively describe the mechanisms of immune evasion in BTC but could expand on actionable therapeutic strategies to address ICI resistance. For example:
- Combination therapies with stromal-targeting agents (e.g., anti-TAMs, Wnt inhibitors) could remodel the tumor microenvironment and improve immune responses.
- Inducing immunogenic cell death (e.g., via radiotherapy, oncolytic viruses, or certain chemotherapies) could help convert "immune-cold" tumors into "immune-hot" tumors.
- Adaptive T-cell therapies, such as CAR-T or TCR-engineered T-cells, may be explored as a complementary approach to ICIs.
- Subgroup Analysis of Immunogenic Classifications: While the transcriptome-based classifications (e.g., "immune-desert," "myeloid") are well described for intrahepatic cholangiocarcinoma (iCCA), there is limited discussion of whether these classifications apply to other BTC subtypes, such as extrahepatic cholangiocarcinoma (eCCA) or gallbladder carcinoma (GBC). Expanding this analysis to include other subtypes would make the review more comprehensive.
Minor:
- Consistency in Terminology:
Throughout the manuscript, the terms "BTC" and "BTCs" are used interchangeably. To improve clarity and consistency, I recommend selecting and consistently using either "BTC" (singular form as a collective noun) or "BTCs" (plural form). Given the subject matter of the paper, the singular form "BTC" may be preferable to refer to the collective entity unless referring specifically to the heterogeneity of subtypes. A similar recommendation applies to the terms “ICI” and “ICIs. - Inclusion of a Summary Figure:
The paper would greatly benefit from a summary figure that visually encapsulates the key elements of the review. For example, the figure could illustrate: - The immune tumor microenvironment (TIME) of BTCs, highlighting key players such as CAFs, TAMs, TILs, and their roles in immune evasion.
- The transcriptome-based classifications (e.g., "immune-desert," "myeloid," and "immunogenomic" subtypes) and their therapeutic implications.
- Current and emerging treatment paradigms, including chemotherapy, ICIs, and targeted therapies, with relevant biomarkers (e.g., PD-L1 expression, FGFR2 mutations, and IDH1 mutations).
Such a visual aid would not only enhance the paper's readability but also serve as a valuable reference for readers seeking to understand the overall landscape of BTC immunotherapy.
Author Response
Comments and Suggestions for Authors
This paper provides a comprehensive and insightful review of the immune landscape of biliary tract cancers (BTCs), focusing on mechanisms of immune evasion, resistance to immune checkpoint inhibitors (ICIs), and potential therapeutic approaches. It is well-written, thoroughly researched, and logically organized, making a valuable contribution to the field of BTC research. The authors have effectively summarized recent advances, including transcriptome-based classifications, tumor microenvironment characteristics, and emerging treatment paradigms that integrate chemotherapy with immunotherapy.
Major:
- Discussion of Biomarkers: While the paper provides an excellent summary of commonly studied biomarkers for immunotherapy (e.g., PD-L1, TMB, MSI-H), it does not sufficiently address the challenges and limitations of these biomarkers in BTC. For example:
- PD-L1 Expression: The lack of correlation between PD-L1 expression and ICI response in BTC could be elaborated. Issues such as spatial variability of PD-L1 within tumors or the limitations of TAP scoring in capturing the complexity of the tumor immune microenvironment should be discussed. Briefly mentioning emerging approaches, such as spatial transcriptomics or multiplex immunohistochemistry, as potential solutions could add value.
Response: Thank you for your comments which we acknowledge. To address this, we have elaborated on the limitations of PD-L1 as a predictive biomarker including spatial variability of PD-L1 and the limitations of the TAP scoring system. We also briefly mentioned spatial transcriptomics and multiplex IHC as emerging approaches to allow a more comprehensive assessment of PD-L1 considering the spatial and cellular context.
- TMB and MSI-H: The paper could elaborate on why these biomarkers are rare in BTC and discuss alternative genomic or epigenomic features (e.g., chromosomal instability or epigenetic silencing) that may be relevant for predicting response to ICIs.
Response: Thank you for your comments. We have elaborated on why these biomarkers are rare in BTC in that it is primarily driven by specific mutations and that MMR deficiency is rare. We briefly mentioned potential alternative predictive biomarkers including chromosomal instability or epigenetic silencing.
- Therapeutic Strategies to Overcome ICI Resistance: The authors effectively describe the mechanisms of immune evasion in BTC but could expand on actionable therapeutic strategies to address ICI resistance. For example:
- Combination therapies with stromal-targeting agents (e.g., anti-TAMs, Wnt inhibitors) could remodel the tumor microenvironment and improve immune responses.
- Inducing immunogenic cell death (e.g., via radiotherapy, oncolytic viruses, or certain chemotherapies) could help convert "immune-cold" tumors into "immune-hot" tumors.
- Adaptive T-cell therapies, such as CAR-T or TCR-engineered T-cells, may be explored as a complementary approach to ICIs.
Response: Thank you for your comments. I have created a new section on therapeutic strategies to overcome ICI resistance discussing stromal targeted agents, immunogenic cell death treatments, adaptive t-cell therapies and a brief section of vaccines and microbiome. This can be found in the revised manuscript section 7.
- Subgroup Analysis of Immunogenic Classifications: While the transcriptome-based classifications (e.g., "immune-desert," "myeloid") are well described for intrahepatic cholangiocarcinoma (iCCA), there is limited discussion of whether these classifications apply to other BTC subtypes, such as extrahepatic cholangiocarcinoma (eCCA) or gallbladder carcinoma (GBC). Expanding this analysis to include other subtypes would make the review more comprehensive.
Response: Thank you for your comments. We have acknowledged the current limitation in data regarding the immune landscape and classification for the eCCA and BTC subtypes and highlighted the need for further studies in the area. We have included the following in the article.
“There is limited data available on how these classifications apply to eCCA and GBC. Research on eCCA has found that this subtype often exhibits a higher degree of immune suppression, characterised by low T-cell infiltration and high levels of immunosuppressive cells such as Tregs[56]. Similarly, GBC has been shown to have a more complex TIME with higher levels of inflammatory cytokines and immune cell infiltration[57]. Despite these findings, comprehensive immune classifications specific to eCCA and GBC have not yet been established, and the immune features of these subtypes remain less well-defined compared to iCCA.”
Minor:
- Comments 1: Consistency in Terminology:
Throughout the manuscript, the terms "BTC" and "BTCs" are used interchangeably. To improve clarity and consistency, I recommend selecting and consistently using either "BTC" (singular form as a collective noun) or "BTCs" (plural form). Given the subject matter of the paper, the singular form "BTC" may be preferable to refer to the collective entity unless referring specifically to the heterogeneity of subtypes. A similar recommendation applies to the terms “ICI” and “ICIs.
Response: Thank you for pointing this out. We have improved the consistency and used the singular form “BTC” and “ICI”.
- Inclusion of a Summary Figure:
The paper would greatly benefit from a summary figure that visually encapsulates the key elements of the review. For example, the figure could illustrate: - The immune tumor microenvironment (TIME) of BTCs, highlighting key players such as CAFs, TAMs, TILs, and their roles in immune evasion.
- The transcriptome-based classifications (e.g., "immune-desert," "myeloid," and "immunogenomic" subtypes) and their therapeutic implications.
- Current and emerging treatment paradigms, including chemotherapy, ICIs, and targeted therapies, with relevant biomarkers (e.g., PD-L1 expression, FGFR2 mutations, and IDH1 mutations).
Such a visual aid would not only enhance the paper's readability but also serve as a valuable reference for readers seeking to understand the overall landscape of BTC immunotherapy.
Response: Thank you for your comments. We have included a summary figure that encapsulates the critical elements of the BTC TIME including Teff, CAF, Treg, TAM and MDSC. It also includes the targetable mutations and the approved treatments that are available.
Reviewer 2 Report
Comments and Suggestions for Authors Discussions on the tumor microenvironment (TME) can be improved further by dividing the section into sub-sections specifically discussing immune-related cells, tumor cells and tumor stromal cells. It is important to highlight the heterogeneity of the TME. As the authors are specifically focusing on the "immune landscape" as per the manuscript title, it would help including sub-sections dedicated to discussions on role of CD8 + T Cells, CD4 + T Cells, Tregs, macrophages, dendritic cells in the TME. The authors may also add in discussions on other immune populations as well if they wish to. It would be useful to have a schematic to support the discussion. Authors can also incorporate a section 5.2.4 discussing cancer vaccines for CCA. The second half of the manuscript will also benefit from having a schematic wherein the authors can highlight different options/molecular routes that are frequently targeted for treatment of CCA.
Author Response
Comments and Suggestions for Authors
Discussions on the tumor microenvironment (TME) can be improved further by dividing the section into sub-sections specifically discussing immune-related cells, tumor cells and tumor stromal cells. It is important to highlight the heterogeneity of the TME. As the authors are specifically focusing on the "immune landscape" as per the manuscript title, it would help including sub-sections dedicated to discussions on role of CD8 + T Cells, CD4 + T Cells, Tregs, macrophages, dendritic cells in the TME. The authors may also add in discussions on other immune populations as well if they wish to. It would be useful to have a schematic to support the discussion. Authors can also incorporate a section 5.2.4 discussing cancer vaccines for CCA. The second half of the manuscript will also benefit from having a schematic wherein the authors can highlight different options/molecular routes that are frequently targeted for treatment of CCA.
Response: Thank you for your comments. We have made changes to the manuscript to better highlight the heterogeneity of the TIME, particularly across the different subtypes of BTC. We have included a summary figure that discusses the roles of the key immune components of the TIME, including effector T cells, regulatory T cells, macrophages and myeloid derived suppressor cells. We expanded on the roles of these cells in the text under the existing sections. Additionally, we have included a new section (section 7) on “therapeutic strategies to overcome immunotherapy resistance” which briefly discusses the potential role of vaccine in CCA. Our main schematic/figure also incorporates the different molecular targets for CCA treatment.
Round 2
Reviewer 1 Report
Comments and Suggestions for Authors
The authors effectively addressed the reviewers' questions and concerns.
Reviewer 2 Report
Comments and Suggestions for Authors
The authors have satisfactorily addressed my comments and I recommend acceptance of the manuscript in its current form.